# MQPD: An Antioxidant Quinone–Dopamine Hybrid Which Induces Sustained Brain Dopamine Elevation

**DOI:** 10.3390/antiox14121416

**Published:** 2025-11-27

**Authors:** Olga Kulikova, Olga Adaeva, Yulia Timoshina, Denis Abaimov, Olga Muzychuk, Maxim Nesterov, Alexander Lopachev, Rogneda Kazanskaya, Dmitry Demchuk, Victor Semenov, Alexander Latanov, Tatiana Fedorova

**Affiliations:** 1Russian Center of Neurology and Neurosciences, 125367 Moscow, Russia; aoi@ioc.ac.ru (O.A.); july.timoschina@yandex.ru (Y.T.); abaimov@neurology.ru (D.A.); muzychuk@neurology.ru (O.M.); a.lopachev@spbu.ru (A.L.); st059046@student.spbu.ru (R.K.); tnf51@bk.ru (T.F.); 2N. D. Zelinsky Institute of Organic Chemistry RAS, 119991 Moscow, Russia; demchuk@ioc.ac.ru (D.D.); vs@ioc.ac.ru (V.S.); 3School of Biology, Lomonosov Moscow State University, 119991 Moscow, Russia; latanov.msu@gmail.com; 4Bioanalytical Laboratory, Scientific Center of Biomedical Technologies of the Federal Medical and Biological Agency of Russia, Svetlye Gory Village, Moscow Region, 143442 Krasnogorsk, Russia; nesms@scbmt.ru; 5Institute of Translational Biomedicine, Saint Petersburg State University, 199034 St. Petersburg, Russia; 6Research Institute for Brain Development and Peak Performance, Peoples’ Friendship University of Russia (RUDN University), 117198 Moscow, Russia

**Keywords:** dopamine, quinone, coenzyme Q, antioxidant, MQPD, blood–brain barrier, Parkinson’s disease, neuroprotection

## Abstract

Parkinson’s disease (PD) therapy is challenged by the multifactorial nature of neurodegeneration, necessitating an approach combining dopamine replenishment and combating oxidative stress. This study characterizes the neuroprotective potential of MQPD, a novel hybrid molecule containing dopamine with a quinone fragment structurally analogous to coenzyme Q. We evaluated MQPD’s antioxidant capacity in vitro using DPPH radical scavenging and lipid peroxidation assays, its neuroprotective efficacy against mitochondrial toxins (rotenone, paraquat) in neuronal cultures, and its ability to modulate striatal dopamine levels in mice. MQPD demonstrated significant antioxidant activity, reduced reactive oxygen species, and was more effective than dopamine or L-DOPA in mitigating toxin-induced cell death. While MQPD itself showed low brain bioavailability, its administration resulted in a sustained increase in striatal dopamine levels for up to four days. The results indicate that MQPD is a potent neuroprotective agent whose effects are likely mediated by direct antioxidant activity and a long-acting mechanism that stabilizes tissue dopamine levels, offering a promising alternative to current therapies for PD.

## 1. Introduction

Dopamine (DA) is a critical neurotransmitter regulating motor function, cognition, and motivation via several central pathways, including the nigrostriatal pathway. The degeneration of this pathway, characterized by the loss of dopaminergic neurons in the substantia nigra pars compacta and their projections to the striatum, is the primary neuropathological cause of the motor symptoms in Parkinson’s disease (PD) [1,2]. A cornerstone of PD management involves strategies to elevate central DA levels. These include the use of monoamine oxidase-B (MAO-B) inhibitors, DA receptor agonists [3,4], and the administration of the DA precursor, L-3,4-dihydroxyphenylalanine (L-DOPA). L-DOPA, a prodrug that crosses the blood–brain barrier (BBB), is decarboxylated to DA in brain tissue and remains the most effective symptomatic treatment of PD. However, its long-term utility is limited by the eventual development of debilitating motor complications, such as dyskinesias and motor fluctuations [5,6]. These adverse effects are thought to arise, in part, from the inherent toxicity of both L-DOPA and DA, which can undergo auto-oxidation to form reactive quinones [7] and, in the case of L-DOPA, be misincorporated into proteins [8], thereby potentially exacerbating neurodegeneration.

However, the clinical use of L-DOPA continues despite its drawbacks, driving the search for alternative compounds that can elevate brain DA without accelerating neurodegeneration. Strategies such as nanoparticle encapsulation [9] or the creation of lipophilic DA conjugates [10,11,12,13] aim to deliver DA across the BBB. Yet, these approaches do not resolve the fundamental toxicity of DA itself, which arises from its auto-oxidation and its enzymatic metabolism by MAO and other enzymes, generating reactive oxygen species (ROS) like hydrogen peroxide [7]. In conditions of oxidative stress, hydrogen peroxide can react with divalent iron to form the highly destructive hydroxyl radical [14]. Therefore, simply replacing L-DOPA with another DA-delivery system fails to address the core issue of DA-induced oxidative stress [15].

Consequently, research has expanded to include antioxidant compounds capable of mitigating oxidative stress and providing neuroprotection [16,17,18]. A prominent candidate is coenzyme Q_10_ (CoQ_10_, ubiquinone), a vital component of the mitochondrial electron transport chain [19] that also functions as an antioxidant by recycling oxidized tocopherols [20]. Despite demonstrated therapeutic potential in models of neurodegeneration [21], the clinical application of native CoQ_10_ is hindered by its poor bioavailability, a result of its large hydrophobic structure and sensitivity to degradation [22]. To overcome this, modified analogs with improved properties have been developed [23]. A notable example is MitoQ, which conjugates CoQ_10_ to a triphenylphosphonium cation to facilitate mitochondrial targeting. While MitoQ showed efficacy in preclinical studies, including cellular and animal models of PD [24,25], it ultimately failed to demonstrate significant benefits in clinical trials, highlighting the ongoing challenge of translating these approaches into effective human therapies [26].

This manuscript characterizes a novel quinone-dopamine conjugate, N-(3,4-dihydroxyfenetyl)-3-(4-hydroxy-2,5-dimethoxy-3,6-dioxocyclohexa-1,4-dien-1-yl)propanamide (MQPD; patents RU2814111C1, RU2843186C1) (Figure 1), designed to cross the BBB and function as a ROS scavenger. We report on the synthesis of MQPD and evaluate its antioxidant properties and cytotoxicity in neuronal cultures relative to DA and L-DOPA. We also evaluate its neuroprotective properties in conditions of glucose-oxygen deprivation (OGD) and oxidative stress induced by mitochondrial toxins in neuronal cell cultures. Furthermore, we demonstrate its ability to cross the BBB in C57/Black mice and induce an elevation of striatal DA tissue content.

## 2. Materials and Methods

### 2.1. Synthesis of MQPD

For the synthesis of N-(3,4-dihydroxyphenethyl)-3-(4-hydroxy-2,5-dimethoxy-3,6-dioxocyclohexa-1,4-dien-1-yl)propanamide (**1**)—(hereinafter MQPD) we used 6, 7-dihydroxy-5,8-dimethoxychroman-2-one (**2**) [27]. This compound reacts easily with nucleophiles and is convenient for the synthesis of various hybrid molecules containing a hydroquinone moiety. Thus, the reaction between compound **2** and 3,4-bis(benzyloxy)phenethylamine hydrochloride (**3**) in the presence of triethylamine gave protected MQPD (**4**) which was used for the next step after simple purification by filtration through silica gel pad. Finally, hydrogenolysis of the benzyl groups in (**4**) led to compound (**1**) in hydroquinone form. This hydroquinone was readily oxidized by atmospheric oxygen to form the desired MQPD **1** (Figure 1).

To a suspension of 6,7-dihydroxy-5,8-dimethoxychroman-2-one (**2**) (3.753 g, 15.62 mmol) and 3,4-bis(benzyloxy)phenethylamine hydrochloride (**3**) (5.778 g, 15.62 mmol) in 52 mL dioxane Et_3_N (4.4 mL, 3.2 g, 31.2 mmol) was added under argon atmosphere. The purple mixture was stirred under argon atmosphere at room temperature for 72 h until complete conversion of the starting compound (TLC monitoring). Then, the mixture was evaporated to dryness, and 50 mL of water and 30 mL of CH_2_Cl_2_ were added to the ink-colored residue. The purple organic layer was separated, and the aqueous layer was extracted with CH_2_Cl_2_ (2 × 30 mL). The organic phases were combined, dried and evaporated in vacuo to obtain 10.404 g ink-colored oil, containing a mixture of quinone and hydroquinone. This oil was dissolved in 30 mL of CH_2_Cl_2_ and stirred under air atmosphere for 24 h until hydroquinone was oxidized to quinone. Next, the product was purified by column chromatography on 80 g of silica gel, eluting with CH_2_Cl_2_:CH_3_OH (100:2.5) to obtain 5.685 g of compound **4**. Brick solid, 64% yield; m. p. 155–157 °C.

^1^H NMR (300 MHz, CDCl_3_): δ = 2.27 (t, J = 7.7 Hz, 2H, CH_2_), 2.65–2.75 (m, 4H, 2 × CH_2_), 3.42 (q., J = 6.6 Hz, 2H, CH_2_), 4.01 (s, 3H, OCH_3_), 4.06 (s, 3H, OCH_3_), 5.14 (s, 2H, OCH_2_Ph), 5.15 (s, 2H, OCH_2_Ph), 5.59 (t, J = 5.7 Hz, 1H, NH), 6.69 (dd., J = 8.1 Hz, J = 1.9 Hz, 1H, CH_Ar_), 6.79 (d., J = 1.8 Hz, 1H, CH_Ar_), 6.88 (d., J = 8.1 Hz, 1H, CH_Ar_), 7.25–7.50 (m, 10H, 2 × Ph).

In a Schlenk tube, to compound **4** (5.573 g, 9.75 mmol) and 10% Pd/C (0.519 g, 0.488 mmol), 50 mL CH_3_OH was added. The mixture was degassed 3 times and hydrogenated at atmospheric pressure at room temperature for 24 h until complete conversion of the starting material (TLC control). The resulting colorless solution was filtered through a pad of celite and a red solution obtained was stirred in air for 72 h until hydroquinone was oxidized to quinone (^1^H NMR monitoring). After that, the mixture was evaporated to dryness to obtain 3.944 g (100%) of **1**. The product was purified by column chromatography on 72 g of silica gel, eluting with CH_2_Cl_2_:CH_3_OH:AcOH (100:10:1) to obtain 2.650 g of compound 1. Red solid, 69% yield; m. p. 58–60 °C.

^1^H NMR (300 MHz, Acetone-d_6_) δ 2.27 (t, J = 7.9 Hz, 2H, CH_2_), 2.62 (t, J = 7.4 Hz, 2H, CH_2_), 2.67 (t, J = 7.9 Hz, 2H, CH_2_), 3.33 (q., J = 6.5 Hz, 2H, CH_2_), 3.88 (s, 3H, OCH_3_), 3.95 (s, 3H, OCH_3_), 6.53 (dd., J = 8.1 Hz, J = 1.8 Hz, 1H, CH_Ar_), 6.70 (d., J = 1.8 Hz, 1H, CH_Ar_), 6.71 (d., J = 8.1 Hz, 1H, CH_Ar_), 7.14 (br.t, J = 4.6 Hz, 1H, NH), 8.00 (2H, br.s, OH). ^13^C NMR (75 MHz, Acetone-d_6_): δ = 20.5, 35.7, 36.0, 41.9, 60.7, 61.6, 116.1, 116.6, 120.9, 130.8, 132.1, 139.0, 142.9, 144.3, 145.9, 154.2, 172.2, 181.4, 184.3 HRMS (ESI-TOF) *m*/*z* [M + H]+ calcd for C_19_H_22_NO_8_: 392.1340; found: 392.1343 (Appendix A).

### 2.2. Evaluation of Antiradical Activity Using the DPPH Test

Antiradical activity was assessed using the 2,2-di(4-tert-octylphenyl)-1-picrylhydrazyl (DPPH) radical scavenging assay [28]. The assay was performed in two solvent systems: a 1:1 (*v*/*v*) water/ethanol mixture and a 1:1 (*v*/*v*) mixture of 50 mM Tris-HCl buffer (pH 7.4) and methanol. Trolox was used as a positive control standard.

Stock solutions of MQPD, DA, and Trolox were prepared in 95% ethanol; due to poor solubility, L-DOPA was dissolved in 0.1 M hydrochloric acid. Working solutions for the water/ethanol system were prepared by diluting the stocks with water and ethanol. For the Tris/methanol system, compounds were dissolved directly in the Tris buffer/methanol mixture.

Test compounds at final concentrations of 6.25 to 100 µM were added to a DPPH solution (final concentration 10 µM). The decrease in optical density at 590 nm was measured immediately using an Ultraspec 3300 Pro spectrophotometer (Amersham Biosciences, Amersham, UK). Kinetic measurements were recorded for 3 min in the ethanol system and for 1 min in the Tris/methanol system. To minimize pre-measurement oxidation, DA was dissolved immediately prior to use. All results are expressed as the mean percentage decrease in DPPH concentration relative to the initial control value ± standard deviation (SD), with each concentration tested at least four times (*n* = 4).

### 2.3. Assessment of Antioxidant Activity in a Lipid Peroxidation Model Using Iron-Induced Chemiluminescence

The antioxidant activity of MQPD was evaluated in vitro using a model of iron-induced peroxidation in low and very-low-density lipoproteins isolated from healthy human donor serum. The assay measures chemiluminescence kinetics during the oxidation of the lipoprotein suspension by ferrous ions (Fe^2+^). Key parameters recorded were: (1) the initial fast flash (h, mV), indicating the baseline level of lipid hydroperoxides; (2) the lag phase duration (τ, s), representing the resistance to oxidation and intrinsic antioxidant capacity; and (3) the maximum chemiluminescence intensity (H, mV), reflecting the overall oxidizability of the substrate.

The antioxidant effects of MQPD were compared directly to those of L-DOPA and DA. Compounds were tested at final concentrations ranging from 0.005 to 1 mM. Chemiluminescence measurements were conducted on an Luminometer-1251 (LKB, Bromma, Sweden) at 37 °C with continuous stirring. Each sample was measured in at least five replicates, and the average values for the chemiluminescence parameters were calculated. Changes in these parameters were expressed as a percentage relative to control samples that did not contain any test compounds [29].

### 2.4. SH-SY5Y Cell Culture Differentiation into Dopaminergic Neurons

The biological activity of MQPD was investigated using the human neuroblastoma cell line SH-SY5Y (ATCC^®^, Manassas, VA, USA), differentiated into a dopaminergic phenotype. All experiments were performed in at least three independent replicates using cells between passages 5 and 15. Cells were maintained in a 1:1 mixture of DMEM/F-12 medium, supplemented with 10% fetal bovine serum (PAA Laboratories, Pasching, Austria), 1% L-glutamine, and 1% penicillin-streptomycin (PanEco, Moscow, Russia). Cultures were incubated at 37 °C in a humidified atmosphere of 5% CO_2_ (ShelLab, Suzhou, China). The medium was replaced every three days, and cells were subcultured as needed every 6–7 days. To induce dopaminergic differentiation, cells were treated with 10 µM retinoic acid (Sigma, St. Louis, MO, USA) for three days after subculturing. The medium was then replaced with a low-serum (1%) medium containing 10 µM retinoic acid and 75 nM phorbol myristate acetate (PMA, Sigma, USA) for an additional four days [30]. All subsequent experimental procedures were conducted on this differentiated culture.

### 2.5. Primary Rat Cortical Neuron Culture

Primary neuronal cultures were prepared from the cerebral cortex of 18-day-old Wistar rat embryos. The cortical tissue was dissected, meninges and blood vessels were removed, and the tissue was rinsed in Ca^2+^- and Mg^2+^-free Hanks’ solution (PanEco, Russia). The tissue was then enzymatically dissociated by incubation in trypsin-EDTA solution (PanEco, Russia) for 15 min at 37 °C. Trypsin was inactivated using 20% fetal bovine serum (FBS, BioSera, Burgess Hill, UK), and the cells were washed twice with Hanks’ solution. The cell pellet was resuspended in MEM medium (PanEco, Russia) supplemented with 10% FBS and 100 U/mL penicillin-streptomycin, then centrifuged at 300× *g* for 3 min. The resulting cell suspension was seeded onto culture plates pre-coated with 0.1 mg/mL poly-L-ornithine (Sigma, USA) for 12 h, at a density of 1.2 × 10^5^ cells per cm^2^. Cultures were maintained in a humidified incubator at 37 °C with 5% CO_2_. After 24 h, the initial medium was replaced with Neurobasal Medium (Gibco, Grand Island, NY, USA) supplemented with 2% B-27 serum-free supplement, 1% GlutaMAX, and 100 U/mL penicillin-streptomycin. The cultures were maintained for 12–14 days in vitro, with half of the medium being replaced with fresh medium every two days.

### 2.6. Oxygen Glucose Deprivation

Rat cortical neurons were subjected to an in vitro model of ischemia–reperfusion injury involving 4 h of OGD followed by 20 h of reoxygenation. Prior to OGD, the culture medium was replaced with glucose-free artificial cerebrospinal fluid (aCSF) composed of (in mM): 125 NaCl, 26 NaHCO_3_, 4 KCl, 1.25 NaH_2_PO_4_, 1.2 MgCl_2_, and 2 CaCl_2_. Control cells were incubated in aCSF containing 25 mM glucose.

To induce hypoxia, culture plates were placed in a hypoxic chamber (New Brunswick Galaxy 48 R, Eppendorf, Germany) at 1% O_2_, 5% CO_2_, 37 °C, and 90% humidity for 4 h. Control plates were maintained under normoxic conditions (atmospheric O_2_). Following OGD, the aCSF was replaced with fresh Neurobasal medium supplemented with GlutaMAX, penicillin-streptomycin, and antioxidant-free B-27 supplement, and cells were returned to a normoxic incubator for the 20 h reoxygenation period.

MQPD was tested at final concentrations of 50, 100, and 200 µM. It was administered either only during the reoxygenation phase or during both the OGD and reoxygenation phases. Cell viability was assessed 24 h after the onset of OGD using the MTT assay.

### 2.7. Assessing Culture Viability Using the MTT Assay

The cytotoxicity of MQPD, 2,2′-azobis(2-methylpropionamidine) dihydrochloride (AAPH), rotenone, and paraquat was evaluated using the MTT assay. Differentiated SH-SY5Y cells were seeded in 96-well plates at a density of 40,000 cells per well. Rotenone was dissolved in dimethyl sulfoxide (DMSO) and diluted in culture medium to a final concentration of 20 µM, ensuring the DMSO concentration did not exceed 1%. Test compounds were added to the culture medium and incubated with the cells for 24 h, with a minimum of eight replicate wells per treatment group.

To account for MQPD’s intrinsic ability to reduce MTT, background absorbance was measured from control wells containing MQPD in culture medium without cells; these values were subtracted from the corresponding experimental wells. After the 24 h incubation, the medium was replaced with fresh medium containing 0.5 mg/mL MTT and incubated for 3 h. The medium was then carefully aspirated, and the resulting formazan crystals were dissolved in 100 µL of DMSO per well.

Absorbance was measured using a Synergy H1 microplate reader (BioTek Instruments Inc., Winooski, VT, USA). The plates were agitated for 6 min, after which the optical density was read at 570 nm and 660 nm. The reference absorbance at 660 nm was subtracted from the measurement at 570 nm to correct for nonspecific background [31,32]. Viability data are expressed as a percentage of the absorbance measured in untreated control wells.

### 2.8. Measurement of LDH Activity

Lactate dehydrogenase (LDH) activity in the culture medium was quantified as a marker of cytotoxicity using a commercial assay kit (Lactate Dehydrogenase Activity Assay Kit, MAK066, Sigma, USA). The collected medium was transferred to a clean 96-well plate. Absorbance at 450 nm was measured at 5 min intervals using a Synergy H1 microplate spectrophotometer (BioTek Instruments Inc., USA). The reaction was monitored until the absorbance of the most active sample surpassed that of the highest concentration NADH standard (12.5 nmol/well). Results are expressed as the group mean, presented as a percentage of the average value from control wells, which was set to 100%.

### 2.9. Measurement of SOD Activity

Primary rat cortical neurons were seeded in 6-well plates at a density of 2 million cells per well. Following treatment with test substances for the designated experimental duration (minimum *n* = 6 wells per group), cells were lysed. The activity of Cu/Zn-superoxide dismutase (SOD) in the lysates was determined using an adaptation of the method by Nishikimi et al., based on the inhibition of nitroblue tetrazolium (NBT) reduction by the superoxide anion radical [33].

Briefly, the reaction mixture containing sodium pyrophosphate buffer (2 mM, pH 8.3), Na-EDTA (2 mM), NBT (0.5 mM), NADH (1.4 mM), and the cell lysate was pre-incubated at 37 °C for one minute. The reaction was initiated by adding 22.2 µM phenazine methasulfate. The kinetics of diformazan formation, the product of NBT reduction, were monitored by measuring the increase in absorbance at 540 nm at 37 °C using a Synergy H1 plate reader (BioTek Instruments Inc., USA). A blank sample without phenazine methasulfate was included for correction.

SOD activity was calculated using the formula: [(ΔA_540_ sample − ΔA_540_ blank) × 2 × reaction volume (µL)]/[sample volume (µL) × time (min) × protein amount (mg) × dilution]. Activity is expressed in units per minute per mg of protein (U/min/mg), where one unit is defined as the amount of enzyme that inhibits the rate of NBT reduction by 50%.

### 2.10. Measurement of MDA Content

The concentration of malondialdehyde (MDA), a marker of lipid peroxidation, was quantified in the same cell lysates used for the SOD activity assay. The assay was performed using a modified thiobarbituric acid reactive substances (TBARS) method proposed by Mihara M. and Ohkawa H., adapted for a microplate format [34,35].

Briefly, cell lysate was mixed with a solution containing 20% acetic acid, 8.1% sodium dodecyl sulfate (SDS), and 0.8% thiobarbituric acid (TBA). The mixture was heated at 95 °C for 120 min in a thermal cycler (TT-2 “Termit”, DNA Technology, Moscow, Russia) to form the pink-colored MDA-TBA adduct. After incubation, samples were cooled on ice and centrifuged at 3000× *g* for 10 min. The absorbance of the supernatant was measured at 535 nm and 580 nm using a Synergy H1 microplate reader (BioTek Instruments Inc., USA).

MDA content was calculated using the formula: [(A_535_ − A_580_)sample − (A_535_ − A_580_)blank] × 10^9^/(1.56 × 10^5^ × protein amount (mg)), where 1.56 × 10^5^ M^−1^cm^−1^ is the molar extinction coefficient of the MDA-TBA complex. Results are expressed as nmol of MDA per mg of protein.

### 2.11. Determination of ROS in Cell Culture

Differentiated neuronal cultures (primary cortical neurons or SH-SY5Y neuroblastoma) were seeded in 24-well plates. Cells were treated with an oxidative stress inducer (2 mM AAPH or 100 µM paraquat) and MQPD, with a minimum of four replicate wells per group, for the specified experimental duration.

Intracellular ROS levels were quantified using the fluorescent probe 2′,7′-dichlorodihydrofluorescein diacetate (DCFH_2_-DA, Invitrogen, Carlsbad, CA, USA). Thirty minutes before the end of the incubation period, DCFH_2_-DA was added to the culture medium at a final concentration of 10 µM. The dye is taken up by living cells and deacetylated to non-fluorescent DCFH_2_, which is trapped intracellularly. In the presence of ROS, DCFH_2_ is oxidized to the highly fluorescent compound 2′,7′-dichlorofluorescein (DCF).

After dye loading, cells were washed three times with Hank’s solution. DCF fluorescence (λ_ex_ 504 nm/λ_em_ 524 nm) was measured from the bottom of each well using a Synergy H1 plate reader (BioTek Instruments Inc., USA), scanning a 5 × 5 point grid. Subsequently, cells were fixed with 3.7% formaldehyde, permeabilized with 0.2% Triton X-100, and nuclei were counterstained with 300 nM DAPI (λ_ex_ 358 nm/λ_em_ 461 nm). Final DCF fluorescence intensity values for each well were normalized to the corresponding DAPI fluorescence to correct for variations in cell number and are expressed as a percentage of the fluorescence in untreated control wells.

### 2.12. Quantification of DA and Its Metabolites Using HPLC-ECD

The intracellular and extracellular content of DA, noradrenaline (NA), and its metabolite 3,4-dihydroxyphenylacetic acid (DOPAC) was determined in homogenates of mouse striatum and in lysates from human SH-SY5Y neuroblastoma cells differentiated into a dopaminergic phenotype. For the cell culture experiments, cells were seeded in Petri dishes at a density of 8 million cells per dish and differentiated. The culture medium was then completely replaced with fresh medium containing 100 µM MQPD and incubated for a specified duration.


*Sample Preparation for Extracellular (Medium) Analysis:*


The culture medium was purified by centrifugation (15 min, 3000 rpm) and filtered through a 0.22 µm membrane to remove cellular debris. For catecholamine extraction, 500 µL of the prepared medium or a calibration solution was mixed with 50 µL of an internal standard (isoprenaline at 5 µg/mL). Subsequently, 50 µL of a 0.1% potassium metabisulfite solution, 1000 µL of 1 M Tris buffer (pH 8.6), and 25 mg of acidic aluminum oxide adsorbent (activated according to Brockman) were added. The samples were shaken on a vibrating mixer for 5 min, then centrifuged at 3000 rpm for 3 min, and the supernatant was discarded. The adsorbent pellet was washed twice with 1 mL of double-distilled water, with shaking for 1 min each time, followed by centrifugation and careful removal of the wash water. For desorption, 50 µL of 0.1 M HClO_4_ was added to the washed adsorbent, shaken for 1 min, and centrifuged at 2000 rpm for 3 min. A 20 µL aliquot of the resulting supernatant was injected into the chromatograph for measuring DA and metabolite content based on a method proposed by Kudrin V. S. et al. with minor modifications [36].


*Sample Preparation for Intracellular Analysis:*


After incubation, the cell culture was washed three times with Hank’s solution. Cells were then lysed in 100 µL of 0.1 N HClO_4_ containing dihydroxybenzylamine (DHBA, 0.25 nmol/mL) as an internal standard for monoamine quantification. The samples were centrifuged twice at 16,000 rpm for 15 min, and the supernatant was used to determine the intracellular content of DA, NA, and DOPAC.


*Chromatographic Analysis:*


Catecholamine content was determined using high-performance liquid chromatography with electrochemical detection (HPLC-ECD) on a System Gold chromatograph (Beckman Coulter, Palo Alto, CA, USA). The system was equipped with a RECIPE EC 3000 amperometric detector (Munich, Germany), a Rheodyne 7125 injector with a 20 µL loop, and a System Gold 125 pump (Palo Alto, CA, USA). Separation was achieved on an Ultrasphere IP reversed-phase column (5 µm, 150 × 4.6 mm) maintained at a mobile phase flow rate of 1 mL/min (~200 atm pressure). The mobile phase consisted of a 0.1 M citrate-phosphate buffer (pH 3.0) containing 1.1 mM octanesulfonic acid, 0.1 mM EDTA, and 9% acetonitrile. Electrochemical detection was performed using a ClinLab ECD-Cell (Munich, Germany) with a glassy carbon working electrode potential of +0.65 V relative to an Ag/AgCl reference electrode.

Data acquisition and chromatogram processing were performed using the MULTICHROM 1.5 hardware-software complex (AMPERSEND, Moscow, Russia). Quantification was based on the internal standard method, whereby the concentrations of monoamines in the samples were calculated from the ratio of the analyte peak area to the internal standard peak area, relative to a calibrated standard curve. All reagents used were of analytical grade.

### 2.13. Experimental Animals and Tissue Sampling

The study was conducted on 72 male C57BL/6 mice aged 6–8 weeks (weight 20–25 g). Animals were housed in the vivarium of the School of Biology, Lomonosov Moscow State University, under standard controlled conditions: temperature 22 ± 2 °C, humidity 55 ± 5%, and a 12 h light/dark cycle with ad libitum access to food and water. All procedures were performed during daylight hours (9:00–20:00). The housing, care, and experimental protocols strictly adhered to international animal welfare standards, including the EU Directive 2010/63/EU, the European Convention for the Protection of Vertebrate Animals, and the Guidance on the Operation of the Animals (Scientific Procedures) Act 1986. The study was approved by the Biomedical Ethics Commission of the institution of Russian Center of Neurology and Neurosciences (Protocol No. 2-9/24, dated 18 March 2024).

Animals were randomly assigned to experimental groups, with group allocation balanced according to their mean body weight. All personnel conducting experiments with animals were blinded to groups; personnel conducting sample analysis received only coded samples with no group indications. Personnel conducting statistical analysis were informed of the groups. MQPD was dissolved in 0.9% NaCl solution with 0.1% DMSO and administered as a single intraperitoneal injection at a dose of 200 mg/kg body weight. Groups of animals (*n* = 6–9 per time point) were euthanized by guillotine decapitation at the following intervals post-injection: 0 min (*n* = 6), 15 min (*n* = 8), 30 min (*n* = 7), 1 h (*n* = 9), 3 h (*n* = 6), 6 h (*n* = 6), 8 h (*n* = 6), 1 day (*n* = 6), 2 days (*n* = 6), 3 days (*n* = 6), and 4 days (*n* = 6). Blood with 4–6 i.u./mL blood natrium heparin solution and brain tissue (striatum and cerebral cortex) were rapidly collected. Blood samples were centrifuged at 3000 rpm for 10 min to obtain plasma. All samples were flash-frozen in liquid nitrogen and stored at −80 °C until analysis.

To assess MQPD BBB penetration, cerebral cortex samples were homogenized. Proteins were precipitated with 20% trichloroacetic acid (TCA) in a 1:2 (tissue/TCA) ratio, followed by double centrifugation at 14,000 rpm for 20 min. The MQPD content was quantified in the resulting supernatant.

To determine the effect on DA levels, striatal samples were homogenized in 20 volumes of 0.1 N HClO_4_ containing 0.5 nmol/mL dihydroxybenzylamine (DHBA) as an internal standard. Homogenates were centrifuged at 10,000× *g* for 15 min, and the supernatants were analyzed for concentrations of DA, DOPAC, homovanillic acid (HVA), and 3-methoxytyramine (3-MT) using the HPLC-ED method described previously.

### 2.14. Quantification of MQPD by HPLC-MS

The MQPD content in mouse blood plasma and cerebral hemisphere samples was quantified using high-performance liquid chromatography-mass spectrometry (HPLC-MS) (Appendix A). Analysis was performed on an HPLC 1290 Infinity system (Agilent Technologies, Santa Clara, CA, USA) coupled to a Q-TOF 6545XT AdvanceBio mass spectrometer (Agilent Technologies, USA). Chromatographic separation was achieved using an Aeris C8 analytical column (2.1 mm × 50 mm, 1.8 µm particle size; Phenomenex, Torrance, CA, USA) with a mobile phase of 0.1% formic acid in water (A) and 0.1% formic acid in acetonitrile (B) at a flow rate of 0.3 mL/min. A 5 µL sample was injected and separated using a 25 min linear gradient: 0–15% B over 2 min, to 25% B over 3 min, to 32% B over 1 min, to 40% B over 5 min, to 55% B over 4 min, to 95% B over 4.5 min, held for 1.5 min, and returned to 0% B over 2 min. Quantitative determination was performed using Mass Hunter software (v.B.09.00, Agilent, Santa Clara, CA, USA), with pure MQPD (99%) as an external standard for calibration and 3-(4-hydroxy-2,5-dimethoxy-3,6-dioxocyclohexa-1,4-dien-1-yl)-N-methylpropanamide as an internal standard (10 µg/mL). The mass spectrometer operated in positive ionization mode with a DuoJet Stream ESI source. Key parameters were: capillary voltage 3500 V, nozzle voltage 1000 V, desolvation gas flow 12 L/min at 300 °C, and nebulizer pressure 35 psi. Full scan spectra (50–1700 *m*/*z*) were acquired with a collision energy of 20 eV for MS/MS fragmentation using nitrogen as the collision gas. Internal mass calibration was performed using purine (*m*/*z* 121.0509) and an Agilent HP0921 (*m*/*z* 922.0098) standard solution. Compound identification was confirmed by MS fragmentation analysis using MS-DIAL software (v.4.60, RIKEN CSRS, Yokohama, Japan), and all measurements were performed in triplicate.

### 2.15. Evaluating Protein Content

The protein content in all brain tissue pellets and cell lysates was quantified to normalize biochemical data. The pellets, obtained after centrifugation, were solubilized in either 0.1 N or 1 N NaOH. The protein concentration in these solutions was then determined using the Lowry method with the DC Protein Assay Kit (Bio-Rad, Hercules, CA, USA).

### 2.16. Statistical Analysis

All in vitro experiments were performed with a minimum of three independent replicates per experimental group. Statistical analysis was conducted using GraphPad Prism software, version 8.0 (GraphPad Software Inc., Solana Beach, CA, USA). Assumptions of normality and homogeneity of variances were assessed using Shapiro–Wilk test and Bartlett’s test. The significance of differences between groups was determined using one-way analysis of variance (ANOVA) followed by Dunnett’s post hoc test for multiple comparisons for data with normal distribution, results are presented as the mean ± standard deviation (m ± SEM). For data sets that did not follow a normal distribution, the non-parametric Kruskal–Wallis test was applied and median with interquartile range (Median [Q1; Q3]) were used to describe results. Differences were considered statistically significant at an adjusted *p*-value of less than 0.05 (*p_adj_* < 0.05).

## 3. Results

### 3.1. Evaluation of MQPD Activity Against the DPPH Radical

We compared the antiradical activity of MQPD, DA and L-DOPA. In both ethanol/water and methanol/water systems, all tested compounds demonstrated high antiradical activity. At high concentrations where the compounds were in stoichiometric excess relative to the DPPH radical (100 and 50 µM in ethanol/water; 50 and 25 µM in methanol/water), antiradical activity was uniformly high and did not differ significantly between compounds, making comparative assessment conditional at these concentrations (Figure 2).

In the ethanol/water system (Figure 2A), the antiradical activity of MQPD at concentrations of 100, 50, and 25 µM was slightly lower than Trolox (by 2.7%, 1.4%, and 9.4%, respectively; *p_adj_* < 0.001), while at 12.5 µM it was equivalent. Compared to DA, MQPD’s activity was slightly higher at 100 and 50 µM but 2.4 and 1.5 times lower at 25 and 12.5 µM, respectively. DA itself exhibited the highest activity, rapidly neutralizing the radical at all concentrations. The activity of L-DOPA could not be evaluated in this system due to its insolubility in ethanol.

In the methanol/water system (Figure 2B), MQPD at 50 µM was 4.7% less effective than Trolox (*p_adj_* < 0.0001), but this difference diminished with decreasing concentration. At 12.5 µM their activities were comparable, and at 6.25 µM MQPD’s activity was 4% higher than Trolox’s (*p_adj_* < 0.05). In contrast, MQPD showed significantly lower activity than both DA and L-DOPA at lower concentrations (12.5 and 6.25 µM).

In summary, the antiradical activity of MQPD was consistently comparable to the standard antioxidant Trolox across both assay systems, while DA demonstrated the highest potency among the compounds tested.

### 3.2. Determination of MQPD Antioxidant Activity by Iron-Induced Chemiluminescence

The antioxidant activity of MQPD was further assessed using a model of Fe^2+^-induced peroxidation in lipoproteins isolated from human donor serum, with results compared to DA and its precursor L-DOPA (Table 1, Appendix A). As shown in Table 1A, MQPD demonstrated a potent capacity to inhibit lipid oxidation across all tested concentrations (0.005–1 mM). It significantly reduced the amplitude of the initial “fast flash” of chemiluminescence (h, mV), which corresponds to the baseline level of lipid hydroperoxides, to a greater extent than both DA and L-DOPA. This reduction indicates MQPD’s ability to neutralize toxic lipid peroxidation products. In contrast, the addition of L-DOPA and DA at the lower concentrations of 0.01 and 0.005 mM did not lead to a significant decrease in hydroperoxide levels relative to the control.

The latent period (τ, s), which reflects the resistance of lipids to oxidation and the direct antioxidant activity of the compound, was also prolonged more effectively by MQPD (Table 1B). The antioxidant activity of both MQPD and DA was significantly higher than the control at concentrations ranging from 0.01 to 1 mM. Notably, MQPD was significantly more effective than both DA and L-DOPA at specific intermediate concentrations of 0.25, 0.1, and 0.05 mM. L-DOPA exhibited the lowest antioxidant activity in this assay, with its effect at concentrations of 0.01 mM and below being indistinguishable from the control.

The maximum chemiluminescence intensity (H, mV), indicative of the total oxidizability of the substrate, was significantly suppressed by all compounds at all concentrations compared to the control (Table 1C). However, at the lowest concentration tested (0.005 mM), MQPD was the most effective, reducing this parameter by 63.6%, compared to a 20.3% reduction by DA (*p_adj_* < 0.0001) and an 11.7% reduction by L-DOPA (*p_adj_* < 0.0001).

At the highest concentrations of 0.5 and 1 mM, all three compounds completely suppressed the development of the lipid peroxidation reaction, making measurement of the latent period and maximum chemiluminescence impossible. A clear, direct concentration-dependent relationship was observed for all chemiluminescence parameters. In summary, while all three compounds exhibited high antioxidant activity, MQPD demonstrated superior efficacy in inhibiting iron-induced lipid peroxidation across most parameters and concentrations.

### 3.3. Determination of MQPD Toxicity for Neuronal Cell Culture

The cytotoxicity of MQPD was evaluated by measuring the viability of human dopaminergic-differentiated SH-SY5Y neuroblastoma cells using the MTT assay, with direct comparison to L-DOPA and DA (Figure 3A–C). The addition of MQPD to the culture medium at concentrations ranging from 50 µM to 500 µM did not alter cell viability, indicating an absence of cytotoxicity within this range. However, at concentrations exceeding 500 µM, a significant, dose-dependent decrease in viability was observed: by 33.8 ± 1.8% at 750 µM (*p_adj_* < 0.001) and by 86.3 ± 0.5% at 1000 µM (*p_adj_* < 0.0001).

Under identical conditions, both DA and L-DOPA demonstrated higher toxicity. DA significantly reduced cell survival starting at a concentration of 50 µM (by 9.6 ± 1.0%, *p_adj_* < 0.0001), while the toxic effect of L-DOPA became significant at 150 µM (32.0 ± 3.8% reduction, *p_adj_* < 0.0001). Therefore, MQPD exhibits substantially lower toxicity than either DA or L-DOPA in this model, with cytotoxic effects manifesting only at concentrations above 500 µM. Furthermore, in a primary rat cortical neuron culture, MQPD showed no toxicity even at the highest tested concentration of 1000 µM (Figure 3D).

### 3.4. Neuroprotective Effect of MQPD Under Induced Oxidative Stress

The neuroprotective efficacy of MQPD was evaluated in a model of generalized oxidative stress induced by adding 2 mM 2,2′-azobis(2-methylpropionamidine) dihydrochloride (AAPH) to cultures of dopaminergic-differentiated SH-SY5Y neuroblastoma cells. AAPH is a potent oxidant that generates ROS in aqueous environments, both intracellularly and extracellularly, initiating oxidation via nucleophilic and free radical mechanisms. MQPD was co-administered with AAPH at concentrations ranging from 10 to 500 µM, and its protective effect was quantified using MTT and LDH assays.

As illustrated in Figure 4A, AAPH significantly reduced cell viability by 21.3 ± 1.8% (*p_adj_* < 0.0001). MQPD demonstrated a concentration-dependent protective effect, increasing viability by 16.9% to 25.9% (*p_adj_* < 0.0001) relative to the AAPH-treated group at concentrations of 100–500 µM. At 75 µM and 50 µM, viability increased by 14.3 ± 4.0% and 14.1 ± 3.7%, respectively (*p_adj_* < 0.001). No significant protective effect was observed at concentrations below 25 µM. These MTT test results indicate that MQPD, at concentrations of 50–500 µM, effectively mitigates the toxic effects of AAPH.

The protective effect was corroborated by measuring LDH activity in the culture medium, a marker of cell membrane integrity and cytotoxicity. MQPD was tested at the most effective concentrations identified in the MTT assay (100–500 µM) (Figure 4B). The addition of 2 mM AAPH increased LDH activity by 13.6 ± 1.7% (*p_adj_* < 0.01). Co-administration of MQPD at all concentrations tested significantly attenuated this increase, reducing LDH activity by 13.6% to 18.5%.

In conclusion, data from both the MTT and LDH assays confirm that MQPD provides a significant protective effect for neuroblastoma cells against AAPH-induced oxidative stress.

### 3.5. Neuroprotective Effect of MQPD Against Rotenone-Induced Oxidative Stress

The efficacy of MQPD was further evaluated in a model of oxidative stress induced by the pesticide rotenone. Rotenone inhibits mitochondrial complex I of the electron transport chain, leading to oxidative stress, ATP depletion, apoptosis, and eventual cell death [37,38].

Incubation of SH-SY5Y cells with 20 µM rotenone for 24 h significantly reduced cell viability to 67.5 ± 1.5% (*p_adj_* < 0.0001), representing a 32.5% decrease compared to untreated control cells (set at 100%, Figure 5A). Co-administration of MQPD with rotenone conferred a concentration-dependent protective effect. Cell viability increased by 15.9 ± 2.3% at 50 µM, 19.2 ± 1.8% at 150 µM, 36.6 ± 2.8% at 250 µM (*p_adj_* < 0.0001), and 34.5 ± 2.0% at 350 µM MQPD relative to the rotenone-only group. The protective effect increased proportionally with MQPD concentration, with viability at 250 and 350 µM MQPD reaching levels indistinguishable from untreated controls. However, increasing the concentration to 500 µM resulted in a diminished protective effect.

Measurement of LDH activity in the culture medium under identical conditions confirmed these findings. Rotenone (20 µM) increased LDH release by 46.4 ± 2.1% (*p_adj_* < 0.0001) compared to intact cells (Figure 5B). MQPD significantly attenuated this increase, reducing LDH activity by 49.4 ± 2.7% at 350 µM and 55.5 ± 3.0% at 250 µM (*p_adj_* < 0.001), and by 60.5% and 57.4% at 150 µM and 50 µM, respectively (*p_adj_* < 0.0001), relative to the rotenone-treated group. The LDH levels in MQPD-treated groups were restored to those of untreated controls.

In conclusion, both the MTT and LDH assays demonstrate that MQPD exhibits significant neuroprotective activity against rotenone-induced toxicity, effectively preventing the associated oxidative stress and cell damage.

### 3.6. Capacity of MQPD to Reduce Intracellular ROS Levels

To elucidate the mechanism underlying its neuroprotective effect, the ability of MQPD to scavenge ROS was investigated in human neuroblastoma cells under oxidative stress induced by 2 mM AAPH. Intracellular ROS levels, measured via DCF fluorescence intensity, were assessed at 15 min, 30 min, 1 h, 2 h, 3 h, and 9 h after treatment (Figure 6).

As shown in Figure 6A, AAPH significantly elevated ROS levels at all time points, ranging from 181.8% to 313.2% of the baseline. Two distinct fluorescence peaks were observed: one at 1 h (281.1 ± 7.1%) and another at 9 h (313.2 ± 7.0%), representing an approximate three-fold increase compared to intact control cells (100 ± 1.4%, *p_adj_* < 0.0001). Co-treatment with MQPD at concentrations of 50, 75, and 100 µM effectively mitigated this increase, reducing ROS levels to those of the intact control at all measured time points. This result demonstrates a potent and sustained direct antioxidant activity of MQPD within the cellular environment.

### 3.7. MQPD Protects Primary Rat Cortical Neurons from Paraquat-Induced Toxicity

The neuroprotective efficacy of MQPD was further assessed on primary rat cortical neurons subjected to oxidative stress induced by the herbicide paraquat. A preliminary dose–response established 100 µM paraquat as the toxic concentration, reducing neuronal viability to approximately 50%. Paraquat toxicity is mediated through inhibition of mitochondrial complexes I and III, leading to oxidative and nitrosative stress [39].

Treatment with 100 µM paraquat significantly decreased neuronal viability to 54.1 ± 1.6% (*p_adj_* < 0.0001) compared to untreated controls. Co-administration of MQPD with paraquat conferred a concentration-dependent protection. MQPD at 25 µM increased viability to 88.1 ± 3.8% (*p_adj_* < 0.0001), and at 100 µM restored viability to levels indistinguishable from intact controls (100.1 ± 1.8%, *p_adj_* < 0.0001; Figure 7A).

Given that paraquat toxicity is ROS-dependent, we evaluated the effect of MQPD on intracellular ROS levels using DCF fluorescence. Paraquat increased ROS levels by 35.7 ± 3.8% (*p_adj_* < 0.001) after one hour of incubation. Co-treatment with 100 µM MQPD completely abrogated this increase, reducing ROS levels by 53.4 ± 6.9% (*p_adj_* < 0.0001) relative to the paraquat-only group, returning them to baseline (Figure 7B).

To investigate the mechanism of protection further, we measured the activity of the key antioxidant enzyme SOD and the levels of MDA, a marker of lipid peroxidation. Incubation with 100 µM paraquat for 30 and 60 min reduced SOD activity by 18.9 ± 6.7% (*p_adj_* < 0.05) and 31.6 ± 0.5% (*p_adj_* < 0.0001), respectively (Figure 8A). While MQPD had no significant effect after 30 min, it increased SOD activity after 60 min by 27.6 ± 3.3% (25 µM, *p_adj_* < 0.001) and 36.5 ± 6.9% (100 µM, *p_adj_* < 0.0001), restoring it to intact levels.

Concurrently, paraquat increased MDA content by 22.1 ± 6.5% (*p_adj_* < 0.05) at 30 min and 30.0 ± 2.7% (*p_adj_* < 0.001) at 60 min (Figure 8B). MQPD effectively reduced MDA levels in a concentration- and time-dependent manner. After 30 min, 25 µM and 100 µM MQPD reduced MDA by 31.0 ± 3.4% (*p_adj_* < 0.01) and 69.2 ± 4.5% (*p_adj_* < 0.0001), respectively. After 60 min, 100 µM MQPD decreased MDA by 46.4 ± 3.1% (*p_adj_* < 0.001), bringing it back to the level of intact cells.

In summary, MQPD demonstrates potent neuroprotection against paraquat toxicity by directly scavenging ROS, restoring the activity of the antioxidant enzyme SOD, and inhibiting lipid peroxidation, with the most pronounced effects observed at a concentration of 100 µM.

### 3.8. MQPD Protects Primary Rat Cortical Neurons from Oxygen-Glucose Deprivation

Given the significant role of oxidative stress in ischemia-induced neurodegeneration [40], we evaluated the efficacy of MQPD in protecting primary rat cortical neurons using an in vitro model of experimental ischemia—OGD. MQPD was administered under two conditions: (1) added to the glucose-free aCSF during the 4 h OGD phase and subsequently to the culture medium during the 20 h reoxygenation phase (preincubation protocol), and (2) added solely to the culture medium during the reoxygenation phase (postincubation protocol).

One-way ANOVA revealed a significant detrimental effect of OGD on neuronal viability (F(4, 108) = 84.50 for preincubation; F(4, 108) = 67.62 for postincubation; *p_adj_* < 0.0001). The OGD insult followed by reoxygenation reduced neuron survival by 27.7 ± 0.4% compared to normoxic controls (*p_adj_* < 0.0001; Figure 9).

MQPD administration under both protocols significantly attenuated this cell death. The preincubation protocol proved slightly more effective, fully restoring viability to the level of intact controls. Specifically, during preincubation, MQPD increased survival by 33.3 ± 1.3% at 50 µM, 38.6 ± 3.2% at 100 µM, and 36.8 ± 4.2% at 200 µM (*p_adj_* < 0.0001). Similarly, postincubation with MQPD also conferred a significant, concentration-dependent increase in neuronal viability. These results demonstrate that MQPD provides substantial neuroprotection against ischemic injury in vitro, whether applied prophylactically or as an intervention post-insult.

### 3.9. Influence of MQPD on Extra- and Intracellular DA Levels in Dopaminergic Neurons

Analysis of catecholamine levels revealed that MQPD induces a rapid and substantial increase in extracellular DA. The extracellular DA concentration rose significantly within 15 min of MQPD application and remained elevated for up to 6 h, returning to baseline levels by 24 h. The peak extracellular DA concentration was observed between 15 min and 1 h, exceeding the level in untreated control cells (2.4 ± 1.3 pmol/mL) by more than tenfold (reaching 26.2–31.1 pmol/mL, *p_adj_* < 0.0001) (Figure 10A).

Similarly, the extracellular concentration of NA peaked 15 min after MQPD addition, showing a dramatic 28-fold increase compared to intact cells (*p_adj_* < 0.0001) (Figure 10B). While the NA level subsequently decreased, it remained significantly elevated for up to 48 h.

### 3.10. In Vivo Delivery of MQPD to the Brain

To evaluate the BBB penetration of MQPD in vivo, a single intraperitoneal injection was administered to male C57BL/6 mice. Plasma and brain samples were collected at time points ranging from 0.25 to 96 h post-injection (Figure 11).

The concentration of MQPD in blood plasma peaked at 15 min, reaching 18.3 ± 2.2 µg/mL (Figure 11A). MQPD was successfully detected in the cerebral hemispheres, confirming its ability to cross the BBB. The highest brain concentration was also observed at 15 min (18.7 ± 3.1 ng/mL), which is approximately 1000-fold lower than the concurrent plasma concentration (Figure 11B). The level in the brain decreased rapidly to 5.6 ± 2.3 ng/mL by 30 min. By 3 h post-administration, MQPD was undetectable in the brains of some animals, though trace amounts (0.3 ng/mL) persisted in a subset of animals (2 out of 6) for up to 4 days.

These results demonstrate that MQPD readily crosses the BBB but is rapidly metabolized and cleared, with the majority eliminated from the brain within 3 h.

### 3.11. Ability of MQPD to Increase DA Content in Brain Tissue

Quantification of DA levels in the mouse striatum revealed a significant and sustained increase following MQPD administration. Compared to the baseline level in intact animals (544.6 ± 33.1 pmol/mg protein), DA content was elevated by 2.2-fold (1181.0 ± 122.0 pmol/mg protein) at 8 h post-injection (Figure 12A). This increase peaked at 2.9-fold (1565 ± 128.2 pmol/mg protein) on day 3 and remained significantly elevated on day 4. The concentrations of three major DA metabolites—DOPAC, HVA, and 3-MT—also increased significantly, although these changes became apparent later than the rise in DA itself, with peak levels observed one day after injection.

The concentration of DOPAC increased 4.7-fold (414.6 ± 62.1 pmol/mg protein, versus 87.6 ± 29.6 pmol/mg protein in control, *p_adj_* = 0.0002) on day 1, declining to levels that remained 3-fold and 2.7-fold above baseline on days 3 and 4, respectively (Figure 12B). HVA levels followed a similar pattern, rising 2.7-fold (145.0 ± 14.0 pmol/mg protein, versus 52.9 ± 16.7 pmol/mg protein in control, *p_adj_* = 0.0006) on day 1 and remaining 2.3-fold higher than intact levels on day 4 (Figure 12C). The content of 3-MT was 2.2-fold higher on day 1 (79.0 ± 8.4 pmol/mg protein, versus 34.6 ± 7.6 pmol/mg protein in control, *p_adj_* = 0.0066) and remained elevated for 3 days, returning to a level not significantly different from intact animals by day 4 (Figure 12D, Appendix A).

These results demonstrate that a single administration of MQPD (200 mg/kg) effectively increases striatal DA levels, with a subsequent rise in its metabolic turnover, confirming that MQPD crosses the BBB and exerts a prolonged effect on the dopaminergic system in vivo.

## 4. Discussion

PD is a multifactorial disorder characterized by complex and interconnected pathological cascades, including aberrant protein aggregation, neuroinflammation, iron dyshomeostasis, oxidative stress, and mitochondrial dysfunction [41]. This complexity precludes the development of a single therapeutic agent targeting all pathways simultaneously. Consequently, the current research focus has shifted towards creating multifunctional molecules that can address several key pathologies [42,43].

In response to this need, we synthesized N-(3,4-dihydroxyphenethyl)-3-(4-hydroxy-2,5-dimethoxy-3,6-dioxocyclohexa-1,4-dien-1-yl)propanamide (methoxyquinone-propanamide, MQPD). This compound is a hybrid molecule integrating two components: DA, which can replenish the neurotransmitter deficit, and a quinonic acid fragment structurally analogous to coenzyme Q metabolites, which enhances lipophilicity and intrinsic antioxidant capacity. This single agent has multiple potential mechanisms of action relevant to PD pathogenesis.

Catecholamines are known to possess intrinsic antioxidant activity attributable to the catechol moiety [44,45]. To assess the impact of the added quinone fragment, we evaluated the ability of MQPD to neutralize the DPPH radical in comparison to DA, its precursor L-DOPA, and the standard antioxidant Trolox. This assay was conducted at a physiologically relevant neutral pH (7.4), as prior research [46] demonstrated that antiradical activity for catecholamines, with DA being the most potent, is higher at neutral pH than under acidic conditions in a water/methanol system.

MQPD exhibited high antiradical activity in both ethanol/water and methanol/water systems, comparable to Trolox. However, its activity profile relative to DA and L-DOPA differed by concentration: while comparable at high concentrations, MQPD’s activity was significantly lower at low concentrations. This divergence can be explained by their distinct oxidation pathways. DA oxidation proceeds through an *o*-quinone intermediate that cyclizes to a leucoaminochrome, a process driven by the nucleophilic free amino group [47,48]. The leucoaminochrome is then oxidized to an aminochrome. In contrast, for MQPD, the amino group is non-nucleophilic and prevents cyclization beyond the initial *o*-quinone stage. This results in a lower overall oxidation capacity for MQPD and, consequently, lower antiradical activity at lower concentrations compared to DA (Figure 13).

In the pathological conditions of PD, the normal oxidative metabolism of DA itself can become a source of toxicity, as enzymes like MAO, tyrosine hydroxylase, and L-amino acid oxidase generate hydrogen peroxide, reactive quinones, and aldehydes as byproducts [7]. In an environment of antioxidant deficiency, hydrogen peroxide can interact with divalent iron via the Fenton reaction to yield highly cytotoxic hydroxyl radicals, exacerbating neuronal damage [14]. In this context, the superior performance of MQPD over DA and L-DOPA is significant. MQPD demonstrated greater efficacy in preventing iron-induced lipid peroxidation in vitro and was more effective at reducing cell death induced by the pro-oxidants AAPH, rotenone, and paraquat—mitochondrial toxins commonly used to model PD [49,50]. Importantly, MQPD exhibited substantially lower inherent toxicity than its parent compounds. Despite this promising initial profile, future studies must rigorously evaluate its long-term safety in animal models, specifically investigating potential off-target effects and the risks associated with chronic quinone accumulation.

The neuroprotective mechanism of MQPD likely involves its mitigation of oxidative damage, as evidenced by the observed reduction in lipid peroxidation and MDA content, coupled with an increase in SOD activity. Additionally, the presence of the quinone fragment in MQPD suggests a direct beneficial effect on mitochondrial function, which may explain its ability to counteract toxins like rotenone and paraquat. This is consistent with the known actions of other quinone-based compounds; for instance, idebenone (a synthetic antioxidant in the coenzyme Q family) supports mitochondrial energy metabolism [51], and pyrroloquinoline quinone protects against rotenone-induced cytotoxicity by preserving mitochondrial morphology and regulating genes critical for mitochondrial biogenesis (PGC-1alpha and TFAM) and dynamics (Drp1 and Mfn2) [52]. The specific impact of MQPD on mitochondria warrants further investigation.

An interesting observation was that MQPD concentrations exceeding 350 μM were less effective in rotenone-induced toxicity models. This may be attributable to rotenone’s known effect of increasing MAO-B activity [53]. At high concentrations, the DA moiety of MQPD could potentially be metabolized by elevated MAO-B, leading to increased hydrogen peroxide production and paradoxically contributing to oxidative stress, thereby diminishing the compound’s protective efficacy.

We next focused on the capacity of MQPD to cross the BBB and elevate DA levels in brain tissue. The lipophilic quinone moiety was designed to enhance BBB penetration and neuronal uptake. While some evidence indicates that supplementary CoQ10 can cross the BBB in animals [54], its derivatives, such as idebenone, exhibit superior penetration [55]. We therefore initially hypothesized that following cellular uptake, the amide bond in MQPD would be cleaved by enzymatic systems, releasing free DA to replenish neuronal stores. A precedent for this prodrug-like mechanism exists with other conjugates, such as an oxazepam-DA molecule, which induced a rapid and significant increase in striatal DA outflow in rats [56].

However, pharmacokinetic data revealed low brain tissue availability of MQPD, along with a correspondingly low direct yield of DA. This finding suggests that the observed increase in DA levels is not primarily due to the direct cleavage of MQPD. Instead, the effect is more likely mediated by indirect mechanisms. These may include the compound’s potent direct antioxidant activity, its overall neuroprotective effect preserving dopaminergic neurons, or a potential inhibition of catecholamine catabolic enzymes. These actions would collectively promote a sustained increase in endogenous tissue DA, contrasting with the acute spike associated with L-DOPA [57].

Conventional levodopa therapy is associated with significant limitations, primarily motor complications and dyskinesias. These arise from the pulsatile, non-physiological stimulation of DA receptors caused by fluctuating plasma levodopa levels, a stark contrast to the continuous stimulation present in the healthy brain [58]. The current standard of care combats the peripheral degradation of levodopa by co-administering a dopa decarboxylase inhibitor (e.g., carbidopa or benserazide), which cannot pass through the BBB itself and minimizes peripheral conversion to DA, reducing side effects and enhancing central delivery [59,60].

In this context, the pharmacokinetic profile of MQPD presents an intriguing alternative. Despite its low and transient direct detection in the brain, MQPD administration results in a delayed yet sustained increase in DA levels, which remain elevated for up to four days. This disconnect between the parent compound’s clearance and the prolonged neurochemical effect suggests a complex mechanism of action. NA has been shown to upregulate TH expression, thereby enhancing DA synthesis at later time points [61]. Thus, the increase in DA after 24 h in vivo may be explained by the MQPD-induced increase in NA content observed in vitro, which could cause an increase in endogenous DA synthesis. Other possible mechanisms could include prolonged stability of its metabolic intermediates, as the sustained presence of drug conjugates has been demonstrated with analogous molecules, such as a lipoic acid-dopamine conjugate, which exhibited prolonged stability in plasma of both rats and humans in vitro [62]. However, further research is needed to clarify the mechanisms behind this observed DA increase.

Therefore, the unique pharmacokinetic profile of MQPD could potentially circumvent the undesirable fluctuations characteristic of levodopa therapy, representing a promising strategy for maintaining stable dopamine levels in the brain. This warrants a detailed investigation into its central metabolism and efficacy in restoring dopaminergic function within established animal models of PD.

Furthermore, our approach aligns with a new paradigm in CNS drug delivery, which recognizes that neurodegenerative diseases themselves can alter BBB properties. For instance, neuroinflammation in PD has been shown to increase BBB permeability through the release of inflammatory mediators [63]. Consequently, it is plausible that the brain bioavailability of MQPD may be enhanced under disease conditions, a hypothesis that requires experimental validation in relevant pathological models.

## 5. Conclusions

MQPD is a novel hybrid molecule rationally designed by conjugating DA, a natural neurotransmitter and receptor agonist, with quinonic acid, a structural analog of ubiquinone metabolites. This design enhances the molecule’s lipophilicity and intrinsic antioxidant potential. In vitro experiments revealed that MQPD exhibits superior antioxidant efficacy and lower neuronal toxicity compared to both DA and L-DOPA. Its mechanisms of action include the direct reduction of ROS and hydroxyl radicals, inhibition of lipid peroxidation, and the enhancement of SOD activity and total antioxidant capacity under oxidative stress. Furthermore, MQPD functions as a coenzyme Q_10_ analogue, mitigating mitochondrial dysfunction. Critically, MQPD administration elevates DA levels in both cell culture models and animal brain tissue following systemic administration.

These findings collectively indicate that MQPD is a promising candidate for further development as a low-toxicity, multifunctional therapeutic agent for PD and other disorders characterized by dopaminergic deficit and oxidative stress. The data presented here provide a strong rationale for future investigations into the efficacy of MQPD in animal models of PD, alongside detailed studies of its mitochondrial effects and metabolization.

## Data Availability

The original contributions presented in this study are included in the article/Appendix A. Further inquiries can be directed to the corresponding author.

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
