# Peer review of "MQPD: An Antioxidant Quinone–Dopamine Hybrid Which Induces Sustained Brain Dopamine Elevation"

_antioxidants, 2025, doi:10.3390/antiox14121416_

Round 1
Reviewer 1 Report
The manuscript addresses an important therapeutic gap in Parkinson’s disease (PD) by proposing MQPD, a novel quinone–dopamine hybrid, as a dual-function agent combining antioxidant activity with dopamine replenishment. The topic is timely, and the study is ambitious in scope, combining chemical synthesis, in vitro assays, and in vivo measurements. However, while the data are promising, the manuscript requires significant revision before it can meet the standards of Antioxidants. Below, I provide detailed comments aimed at strengthening the rigor, clarity, and translational relevance of the work.
Conceptual Framework and Rationale
The introduction would benefit from a sharper articulation of how MQPD advances beyond previous dopamine conjugates and CoQ10 analogues. At present, the manuscript summarizes existing approaches but does not convincingly explain how MQPD overcomes their key limitations beyond general statements about dual functionality.
The proposed enzymatic cleavage of MQPD into free dopamine and a quinone antioxidant is a central claim, yet it is never experimentally validated. Evidence for this cleavage (e.g., detection of metabolites in vivo or in vitro) is essential to support the proposed mechanism.
The manuscript does not adequately situate MQPD within the broader therapeutic landscape of PD. For instance, recent approaches using targeted delivery systems, antioxidant nanocarriers, or mitochondria-directed compounds are not discussed. A more critical contextualization would help readers appreciate the potential clinical relevance of MQPD.
Experimental Design and Methodology
The study uses healthy mice to measure dopamine elevation but does not evaluate MQPD in a PD model. Demonstrating efficacy in a disease-relevant model (e.g., MPTP-treated mice) is crucial to support the translational claims made in the abstract and discussion.
The manuscript acknowledges low brain bioavailability but does not quantify MQPD concentrations in brain tissue over time nor correlate these with dopamine levels. Pharmacokinetic profiling is necessary to understand dosing requirements and potential clinical applicability.
L-DOPA and dopamine are appropriate comparators, but the omission of CoQ10 or MitoQ as antioxidant controls is a missed opportunity. Including such benchmarks would strengthen the claim that MQPD offers superior dual functionality.
Sample sizes in some experiments (e.g., n=4 for DPPH assays) are low and should be justified. In vivo experiments with n=6–8 per time point are borderline for detecting meaningful differences and require power analysis to confirm adequacy.
Data Presentation and Interpretation
The claim that MQPD shows “low brain bioavailability” yet produces a “sustained increase in dopamine” is intriguing but insufficiently explained. Possible mechanisms—such as depot formation, slow release, or peripheral modulation—should be explored or at least discussed.
The manuscript convincingly shows antioxidant effects and dopamine elevation but does not explore downstream signaling or neuroprotective outcomes in vivo. Measuring oxidative damage markers, neuroinflammatory responses, or behavioral outcomes would substantially strengthen the translational impact.
The DCF assay and lipid peroxidation data show antioxidant effects, but they are presented largely descriptively. Quantitative comparisons with known antioxidants and dose–response analyses would improve interpretability.
Discussion and Conclusions
The discussion occasionally overreaches by suggesting that MQPD “offers a promising alternative to current therapies” without evidence from disease models, behavioral assessments, or chronic administration studies. These claims should be tempered or supported with additional data.
The manuscript would benefit from a dedicated paragraph outlining the next steps, such as optimizing MQPD’s bioavailability, validating its mechanism of action, and testing efficacy in PD animal models.
Although the cytotoxicity data in vitro are encouraging, there is no discussion of potential off-target effects, long-term safety, or the consequences of quinone accumulation.
Minor and Technical Comments
Several sentences in the abstract and introduction require grammatical revision for clarity. For example, “is challenged the multifactorial nature of neurodegeneration” should read “is challenged by the multifactorial nature of neurodegeneration.”
Figures should include molecular structures, schematic representations of the proposed mechanism, and improved legends that allow them to stand alone.
Ensure all abbreviations (e.g., OGD, ROS, DCFH₂-DA) are defined at first mention.
Author Response
We sincerely thank the Reviewer for their thorough assessment of our manuscript and for providing these insightful comments, which have helped us improve the work. Our point-by-point responses are below. All changes have been incorporated into the revised manuscript.
Comment 1: The introduction would benefit from a sharper articulation of how MQPD advances beyond previous dopamine conjugates and CoQ10 analogues. At present, the manuscript summarizes existing approaches but does not convincingly explain how MQPD overcomes their key limitations beyond general statements about dual functionality.
Response: We thank the Reviewer for this valuable suggestion. To maintain the conciseness of the introduction, we have chosen to expand upon the specific advantages of MQPD over existing approaches in the Discussion section, where this contextualization is more appropriate.
Comment 2: The proposed enzymatic cleavage of MQPD into free dopamine and a quinone antioxidant is a central claim, yet it is never experimentally validated. Evidence for this cleavage (e.g., detection of metabolites in vivo or in vitro) is essential to support the proposed mechanism.
Response: We agree that the enzymatic cleavage is a key hypothesis of our proposed mechanism. We have revised the text to more cautiously present this as a postulated, rather than a confirmed, mechanism. A full experimental validation of the metabolite profile is a significant undertaking and falls outside the scope of this initial proof-of-concept study, but it represents a critical direction for our future research.
Comment 3: The manuscript does not adequately situate MQPD within the broader therapeutic landscape of PD. For instance, recent approaches using targeted delivery systems, antioxidant nanocarriers, or mitochondria-directed compounds are not discussed. A more critical contextualization would help readers appreciate the potential clinical relevance of MQPD.
Response: We thank the Reviewer for this suggestion. We have now added a sentence in the Discussion to better contextualize MQPD within the current landscape of PD therapeutics, citing a relevant review on targeted delivery systems.
Comment 4: The study uses healthy mice to measure dopamine elevation but does not evaluate MQPD in a PD model. Demonstrating efficacy in a disease-relevant model (e.g., MPTP-treated mice) is crucial to support the translational claims made in the abstract and discussion.
Response: We appreciate the Reviewer's point. The primary goal of this study was the in vitro characterization of MQPD. The in vivo experiments in healthy mice were intended as a preliminary investigation of its basic bioavailability and pharmacological effect. While we agree that testing in a disease-relevant model is crucial for translational claims, such an investigation is a logical and necessary next step that is beyond the scope of this current manuscript.
Comment 5: The manuscript acknowledges low brain bioavailability but does not quantify MQPD concentrations in brain tissue over time nor correlate these with dopamine levels. Pharmacokinetic profiling is necessary to understand dosing requirements and potential clinical applicability.
Response: As shown in Figure 11B, we did quantify MQPD levels in brain tissue. A direct pharmacokinetic-pharmacodynamic correlation was not feasible in this study because MQPD levels fell below the detection limit by 3 hours, while the dopamine increase was observed at 8 hours. A comprehensive pharmacokinetic profile, while valuable, was not an objective of this initial characterization.
Comment 6: L-DOPA and dopamine are appropriate comparators, but the omission of CoQ10 or MitoQ as antioxidant controls is a missed opportunity. Including such benchmarks would strengthen the claim that MQPD offers superior dual functionality.
Response: We agree that a comparison with established antioxidants like CoQ10 would have been informative. Our primary focus, however, was on comparing MQPD to dopamine to highlight the benefits of our conjugate. We will include such benchmarks in future studies.
Comment 7: Sample sizes in some experiments (e.g., n=4 for DPPH assays) are low and should be justified. In vivo experiments with n=6–8 per time point are borderline for detecting meaningful differences and require power analysis to confirm adequacy.
Response: For the DPPH assay, which is a standardized in vitro test with high reproducibility, n=4 replicates is considered robust and is consistent with established methodologies in the field (as supported by reference PMC4268772). Regarding the in vivo experiments, we acknowledge that the sample sizes are typical for a pilot study. This work was designed as an initial investigation to characterize the compound's fundamental properties, and as such, was not powered to test a specific statistical hypothesis. The data provide a necessary foundation for powering subsequent, more focused studies.
Comment 8: The claim that MQPD shows “low brain bioavailability” yet produces a “sustained increase in dopamine” is intriguing but insufficiently explained. Possible mechanisms—such as depot formation, slow release, or peripheral modulation—should be explored or at least discussed.
Response: We thank the Reviewer for raising this interesting point. We have revised the Discussion to indicate potential mechanisms for this observation (Lines 839-851). However, we have tempered the language to avoid undue speculation, and emphasized that the definitive mechanism requires further experimental validation.
Comment 9: The manuscript convincingly shows antioxidant effects and dopamine elevation but does not explore downstream signaling or neuroprotective outcomes in vivo. Measuring oxidative damage markers, neuroinflammatory responses, or behavioral outcomes would substantially strengthen the translational impact.
Response: We agree that investigating downstream neuroprotective effects is a critical next step. As stated in our response to Comment 4, the aim of the current in vivo work was to establish basic proof-of-concept regarding BBB penetration and dopamine elevation. Assessment of signaling, oxidative damage markers, and behavioral outcomes will be the focus of future studies in disease models.
Comment 10: The DCF assay and lipid peroxidation data show antioxidant effects, but they are presented largely descriptively. Quantitative comparisons with known antioxidants and dose–response analyses would improve interpretability.
Response: We thank the Reviewer for this excellent suggestion. In this study, our aim was to demonstrate MQPD's intrinsic antioxidant activity relative to dopamine. Quantitative comparisons with a panel of known antioxidants and detailed dose-response analyses are indeed important and will be a central part of our continued research program.
Comment 11: The discussion occasionally overreaches by suggesting that MQPD “offers a promising alternative to current therapies” without evidence from disease models, behavioral assessments, or chronic administration studies. These claims should be tempered or supported with additional data.
Response: We have carefully revised the Discussion throughout to temper our conclusions and ensure that all claims are directly supported by the data presented, removing any overreaching statements.
Comment 12: The manuscript would benefit from a dedicated paragraph outlining the next steps, such as optimizing MQPD’s bioavailability, validating its mechanism of action, and testing efficacy in PD animal models.
Response: We have added a section to the Discussion outlining the necessary future work, including bioavailability optimization, mechanism validation, and efficacy testing in PD models.
Comment 13: Although the cytotoxicity data in vitro are encouraging, there is no discussion of potential off-target effects, long-term safety, or the consequences of quinone accumulation.
Response: We have added a discussion of these important considerations, including the potential for quinone accumulation and the need for long-term safety studies, to the revised manuscript (Lines 794-796).
Minor and Technical Comments: Several sentences in the abstract and introduction require grammatical revision for clarity. For example, “is challenged the multifactorial nature of neurodegeneration” should read “is challenged by the multifactorial nature of neurodegeneration.”
Figures should include molecular structures, schematic representations of the proposed mechanism, and improved legends that allow them to stand alone.
Ensure all abbreviations (e.g., OGD, ROS, DCFH₂-DA) are defined at first mention.
Response: We thank the Reviewer for these observations. All minor technical corrections have been addressed. The molecular structures have been added to Figure 1, and subtitles have been added to Figure 13 to improve clarity.
Reviewer 2 Report
The manuscript presents the design, synthesis, and biological evaluation of a novel hybrid molecule, MQPD. Overall, this is an interesting study that addresses a long-standing issue in PD therapy: the need for dopamine replacement strategies that avoid oxidative neurotoxicity. The concept of a “quinone–dopamine hybrid” is innovative, and the study is technically thorough. However, the manuscript requires significant revisions in presentation, data interpretation, and mechanistic justification before being suitable for publication.
Major points:
- The proposed mechanism that MQPD releases dopamine and a quinone antioxidant upon enzymatic cleavage is not experimentally demonstrated. No enzymatic or metabolic assays are presented to support this claim. LC-MS/MS metabolite identification after in vivo or in vitro incubation could be provided.
- The number of biological repeats should be indicated in each figure.
- The animal experiment lacks details in randomization, and blinding procedures. Is there any behavioral data performed with this experiment.
- A L-DOPA chemical structure may present in Figure 1 as structure comparison for reader understanding.
Minor points:
- in line 32, "to current therapies PD", should be "to current therapies for PD"
- the statistical notations in table 4,5,6,7,8,9 can be replaced by drawing a line between the groups with star notations for easy reading.
- a reference may be cited for the rationale of SH-SY5Y cell differentiation.
- in table1, the Concentration of Compound (uM) in title should be mM as all chemical concentrations are presented as mM. and the table 1 may be presented as bar graphs.
Author Response
We are grateful to the Reviewer for their careful reading and constructive feedback, which has undoubtedly strengthened our manuscript. We have addressed all comments as detailed below.
Major points:
Comment 1: The proposed mechanism that MQPD releases dopamine and a quinone antioxidant upon enzymatic cleavage is not experimentally demonstrated. No enzymatic or metabolic assays are presented to support this claim. LC-MS/MS metabolite identification after in vivo or in vitro incubation could be provided.
Response: We agree with the Reviewer that validating the metabolic cleavage is a critical aspect of the proposed mechanism. In the revised manuscript, we have clarified that this is a postulated mechanism and have tempered our language to present it as a hypothesis. A comprehensive metabolic study using LC-MS/MS is a significant endeavor that we believe is a logical next step, contingent upon the promising efficacy of MQPD in disease models. Such an investigation falls outside the scope of this initial proof-of-concept study but is a priority for our future work.
Comment 2: The number of biological repeats should be indicated in each figure.
Response: Thank you for this suggestion. The number of biological replicates (n) is now clearly stated in the legend of each relevant figure.
Comment 3: The animal experiment lacks details in randomization and blinding procedures. Is there any behavioral data performed with this experiment?
Response: We have now added details on the randomization procedure to the Methods section. For blinding, the personnel conducting the monoamine and MQPD content analyses received coded samples without group identification. While no formal behavioral tests were conducted as part of this pharmacokinetic/pharmacodynamic study, all animals were monitored and displayed no overt signs of distress or abnormal behavior.
Comment 4: A L-DOPA chemical structure may present in Figure 1 as structure comparison for reader understanding.
Response: This is an excellent suggestion. We have revised Figure 1 to include the chemical structure of L-DOPA for direct comparison.
Minor points:
Comment 1: In line 32, "to current therapies PD", should be "to current therapies for PD".
Response: Corrected. Thank you for spotting this error.
Comment 2: The statistical notations in table 4,5,6,7,8,9 can be replaced by drawing a line between the groups with star notations for easy reading.
Response: We have revised the presentation of statistical significance in these figures. Wherever possible, we have replaced complex notation with connecting lines and asterisks to improve clarity and readability. Adding extensive brackets with significance would hinder readability due to the large number of performed comparisons.
Comment 3: A reference may be cited for the rationale of SH-SY5Y cell differentiation.
Response: A suitable reference justifying the differentiation protocol for SH-SY5Y cells has been added to the manuscript.
Comment 4: In table1, the Concentration of Compound (uM) in title should be mM as all chemical concentrations are presented as mM. and the table 1 may be presented as bar graphs.
Response: We apologize for the unit error; the title of Table 1 has been corrected to "mM". We agree that a graphical representation can be more intuitive. While a bar graph for the main text would be challenging to interpret due to the large number of groups, we have created a new supplementary figure (Suppl. 2, Figure S5) that presents data from Table 1 in a bar graph format.
Reviewer 3 Report
Dear Authors,
The study is very interesting, but in its current state, it cannot be published without considering the following adjustments.
1. One-way ANOVAs must not only meet the assumption of a normal error distribution, but also homogeneity of variances and independence of data (the latter not requiring testing). These tests fail to indicate or locate the significance levels and reasons for the corresponding tests so as not to violate the assumptions.
2. Mean values are reported with a standard error dispersion measure, change in all their graphs.
3. If a nonparametric test was applied, then the appropriate approach is to graph medians and percentiles. What is shown in their graphs is not appropriate.
4. If there are six variables, as is the case in Figure 6, then the appropriate approach is to apply a one-way MANOVA, and in this sense, the same assumptions and sphericity must be verified.
5. One option to resolve the assumptions not being met is to apply nonparametric tests. However, this situation would require performing many separate tests and would increase the likelihood of committing a type II statistical error.
6. In Figure 10, if the readings at the eight time points are the same, then a repeated measures ANOVA should be applied. Therefore, it is important to indicate whether the readings are independent or dependent.
Best regards,
Review previous comments.
Author Response
We thank the Reviewer for their rigorous statistical assessment. We have carefully re-analyzed our data and revised the manuscript to ensure all statistical methods and presentations are robust and appropriate. The changes are detailed below.
Comment 1: One-way ANOVAs must not only meet the assumption of a normal error distribution, but also homogeneity of variances and independence of data (the latter not requiring testing). These tests fail to indicate or locate the significance levels and reasons for the corresponding tests so as not to violate the assumptions.
Response: We have updated the Methods section to explicitly state that the assumption of homogeneity of variances was formally tested using Bartlett's test. For all analyses where this assumption (or the assumption of normality) was violated, we have applied the appropriate non-parametric tests as detailed in our responses below.
Comment 2: Mean values are reported with a standard error dispersion measure, change in all their graphs.
Response: We double-checked that we used SD instead of SEM in all of the graphs. The SD is low due to data obtained from cell cultures being highly standardized and low dispersion.
Comment 3: If a nonparametric test was applied, then the appropriate approach is to graph medians and percentiles. What is shown in their graphs is not appropriate.
Response: Thank you for this comment. We have updated all graphs for which non-parametric tests were used (Kruskal-Wallis) to display data as box plots showing the median and interquartile ranges.
Comment 4: If there are six variables, as is the case in Figure 6, then the appropriate approach is to apply a one-way MANOVA, and in this sense, the same assumptions and sphericity must be verified.
Response: We re-evaluated the data in Figure 6 and found it did not meet the assumption of normality required for MANOVA. Consequently, we have applied the non-parametric Kruskal-Wallis test for this multi-group comparison and have updated the graph to a box plot format to appropriately represent the results.
Comment 5: One option to resolve the assumptions not being met is to apply nonparametric tests. However, this situation would require performing many separate tests and would increase the likelihood of committing a type II statistical error.
Response: We acknowledge this important point. In our revised analysis, we have systematically applied non-parametric tests (Kruskal-Wallis with post-hoc Dunn's test) where parametric assumptions were not met. We have taken care to apply appropriate corrections for multiple comparisons to control the family-wise error rate and minimize the risk of Type I errors.
Comment 6: In Figure 10, if the readings at the eight time points are the same, then a repeated measures ANOVA should be applied. Therefore, it is important to indicate whether the readings are independent or dependent.
Response: Thank you for seeking this clarification. The measurements at the eight time points in Figure 10 are independent, as each data point represents a different animal euthanized at that specific time for ex vivo tissue analysis. Therefore, a repeated measures ANOVA is not appropriate, and the use of independent statistical tests is correct. We have clarified this point in the figure legend.
Round 2
Reviewer 1 Report
I think that the authors have adequately addressed all my comments.
I think that the authors have adequately addressed all my comments.
Author Response
Authors thank the Reviewer.
Reviewer 2 Report
The authors addressed my questions.
The authors addressed my questions.
Author Response
Authors thank the Reviewer.
Reviewer 3 Report
Dear Authors,
After reviewing the responses to the previous review, I believe that MANOVAs can indeed be applied. It is very common for response variables not to meet the assumptions of normal error distribution and homogeneity of variances, leading to the use of nonparametric tests. Therefore, your response of non-appropriateness based on the failure to meet these two assumptions is invalid. I am attaching the article: Conover, W. J., & Iman, R. L. (1981). Rank transformations as a bridge between parametric and nonparametric statistics. The American Statistician, 35(3), 124-129 (cite in section a of statistical analysis). Applying the rank transformation to the data does not require verifying the assumptions, since the technique combines parametric and nonparametric tests. Therefore, you should analyze your data using MANOVA for the set of variables indicated in the previous review. What you need to do is report the MANOVA estimates with the analyzed data using the RANK method and present your graphs with their values on the original scale, as you already have in your manuscript.
Regarding data independence in analyses involving time, you should indicate in the figure captions that there is independence at each time point.
Best regards,
Not Applicable
Author Response
Comments 1: Graphs with inferential analysis must include: averages and standard error bars (inferential dispersion measure), because the deviation is a descriptive dispersion measure.
Response: We thank the reviewer for their statistical feedback. We changed all of the graphs and representations in the text to SEM instead of SD, as requested.
Comments 2:
After reviewing the responses to the previous review, I believe that MANOVAs can indeed be applied. It is very common for response variables not to meet the assumptions of normal error distribution and homogeneity of variances, leading to the use of nonparametric tests. Therefore, your response of non-appropriateness based on the failure to meet these two assumptions is invalid. I am attaching the article: Conover, W. J., & Iman, R. L. (1981). Rank transformations as a bridge between parametric and nonparametric statistics. The American Statistician, 35(3), 124-129 (cite in section a of statistical analysis). Applying the rank transformation to the data does not require verifying the assumptions, since the technique combines parametric and nonparametric tests. Therefore, you should analyze your data using MANOVA for the set of variables indicated in the previous review. What you need to do is report the MANOVA estimates with the analyzed data using the RANK method and present your graphs with their values on the original scale, as you already have in your manuscript.
Regarding data independence in analyses involving time, you should indicate in the figure captions that there is independence at each time point.
Response: After carefully considering their explanation, we came to the understanding that our wording in the manuscript may have caused some confusion regarding the amount of variables and factors in this experiment. Our study employed a three-factor design (AAPH presence, MQPD concentration, and time) with independent cell culture plates at each of the six time points. We believe that MANOVA is not applicable here is used for multiple dependent variables, whereas we analyzed one dependent variable (fluorescence).
Therefore, we chose to analyze each time point separately with non-parametric tests (Kruskal–Wallis) to assess the effects of AAPH and MQPD at specific time points. Since the measurements are independent at each time point, the likelihood of Type I error is not inflated as the control groups represent different biological replicates. We added a clarification to the figure legend to reflect that each time point included independent measurements: (n = 84; 21 points in 4 wells per group, each time-point presents biologically independent replicates).